# Inhibition of Heparanase Expression Results in Suppression of Invasion, Migration and Adhesion Abilities of Bladder Cancer Cells

**DOI:** 10.3390/ijms21113789

**Published:** 2020-05-27

**Authors:** Yoshihiro Tatsumi, Makito Miyake, Keiji Shimada, Tomomi Fujii, Shunta Hori, Yosuke Morizawa, Yasushi Nakai, Satoshi Anai, Nobumichi Tanaka, Noboru Konishi, Kiyohide Fujimoto

**Affiliations:** 1Department of Urology, Nara Medical University, 840 Shijo-cho, Nara 634-8522, Japan; takuro.birds.nest@gmail.com (Y.T.); makitomiyake@yahoo.co.jp (M.M.); horimaus@gmail.com (S.H.); tigers.yosuke@gmail.com (Y.M.); nakaiyasusiuro@live.jp (Y.N.); sanai@naramed-u.ac.jp (S.A.); sendo@naramed-u.ac.jp (N.T.); 2Department of Pathology, Nara Medical University, 840 Shijo-cho, Nara 634-8522, Japan; k-shimada@nara-jadecom.jp (K.S.); fujiit@naramed-u.ac.jp (T.F.); n-konishi@takai-hp.com (N.K.)

**Keywords:** heparanase, syndecan-1, heparan sulfate proteoglycans (HSPGs), urothelial carcinoma

## Abstract

Heparan sulfate proteoglycan syndecan-1, CD138, is known to be associated with cell proliferation, adhesion, and migration in malignancies. We previously reported that syndecan-1 (CD138) may contribute to urothelial carcinoma cell survival and progression. We investigated the role of heparanase, an enzyme activated by syndecan-1 in human urothelial carcinoma. Using human urothelial cancer cell lines, MGH-U3 and T24, heparanase expression was reduced with siRNA and RK-682, a heparanase inhibitor, to examine changes in cell proliferation activity, induction of apoptosis, invasion ability of cells, and its relationship to autophagy. A bladder cancer development mouse model was treated with RK-682 and the bladder tissues were examined using immunohistochemical analysis for Ki-67, E-cadherin, LC3, and CD31 expressions. Heparanase inhibition suppressed cellular growth by approximately 40% and induced apoptosis. The heparanase inhibitor decreased cell activity in a concentration-dependent manner and suppressed invasion ability by 40%. Inhibition of heparanase was found to suppress autophagy. In N-butyl-N-(4-hydroxybutyl) nitrosamine (BBN)-induced bladder cancer mice, treatment with heparanase inhibitor suppressed the progression of cancer by 40%, compared to controls. Immunohistochemistry analysis showed that heparanase inhibitor suppressed cell growth, and autophagy. In conclusion, heparanase suppresses apoptosis and promotes invasion and autophagy in urothelial cancer.

## 1. Introduction

Heparan sulfate proteoglycans (HSPGs) are glycoproteins containing heparan sulfate (HS) groups that are covalently attached [1]. They are widely expressed and play critical roles in numerous cellular processes, including endocytosis, migration, and adhesion. Their actions are mediated through interactions with ligands such as growth factors, cytokines, extracellular matrix proteins, and enzymes. HSPGs are also involved in the malignant transformation of cells. Growth factors are released through cellular degradation, which promote the invasion and proliferation of cancer cells [2,3]. Syndecan-1 (SDC1) is an HSPG that is overexpressed in both normal and malignant cells, contributing to the development of hematopoietic and carcinoma development [4,5,6,7,8,9]. Syndecan-1 regulates substance permeation and constitutes a reservoir for various growth factors and cytokines in the basement membrane of cells. It plays a critical role in the progression and invasion of urothelial cancer through enhanced angiogenesis.

We have previously reported that syndecan-1 (CD138) suppresses apoptosis and increases the capacity for cell proliferation via junB-FLIP long signal in urothelial cancer [10]. Urothelial cancers can be broadly classified into low-grade, non-invasive, and high-grade invasive cancers [11]. Invasive urothelial cancers exhibit significantly higher syndecan-1 expression. In vitro experiments showed that knocking down syndecan-1 using siRNA induces cellular apoptosis and decreases the capacity for cellular proliferation. The enzyme heparanase controls the activation of syndecan-1.

Heparanase is an endoglycosidase enzyme that targets HSPG proteins expressed in the extracellular matrix (ECM) and basement membrane (BM) for degradation [12]. Heparanase activation expedites the movement of tumor cells through the ECM and BM, facilitating metastasis. Heparanase is also known to be expressed in many types of malignant tumors, and is associated with metastasis and angiogenesis [13,14]. Heparinase cleavage of HSPGs produces soluble proteins that infiltrate into the tumor microenvironment, where they interact with ligands such as growth factors, modifying signaling pathways [15]. SDC1 in the stroma promotes breast carcinoma growth by enhancing FGF2 signaling [16]. Heparanase-neutralizing antibodies have been suggested for the treatment of diffuse non-Hodgkin’s B-cell and follicular lymphomas [17] through the inhibition of cell invasion and tumor metastasis processes [17,18,19]. Recently, a small molecule inhibitor of heparanase was shown to reduce metastatic characteristics in a hepatocellular carcinoma model [20]. In vivo studies using heparanase inhibitors in animal tumor models have also demonstrated reductions in tumor metastasis [21,22,23]. As heparinase is absent or expressed at low levels in normal tissue [24,25], it may be a potential target candidate for therapeutic interventions. Various studies have investigated the underlying mechanism for heparanase activity in cancer, including enhancement of angiogenesis and promotion of apoptosis and autophagy [14,26,27,28,29,30]. Reports indicate that autophagy contributes to chemotherapy resistance development, making this an important research focus area.

In this study, we analyzed the function of heparanase, an activator of syndecan-1, in angiogenesis, apoptosis, and autophagy. The aim was to establish heparanase as a target for molecular therapy in urothelial carcinoma.

## 2. Results

### 2.1. Heparanase Is Overexpressed in Human Urothelial Carcinoma of the Urinary Bladder, and Heparanase Expression Levels Are Associated with Intravesical Recurrence

The patterns of heparanase expression in resected bladder cancer tissue samples were analyzed using immunohistochemical (IHC) staining. The relationship between heparanase expression and recurrence, metastasis, and prognosis of urothelial cancer was examined. Tissue specimens (*n* = 57) were resected from the renal pelvis, ureter, and bladder of patients with multifocal onset, relapse, metastasis, and prognosis cases of urothelial cancer. Table 1 summarizes patient clinicopathological data using the 2009 World Health Organization (WHO) grading and staging of tumors classification [31]. The total number of Ta cases was 20 and the grade was low grade:high grade, 13:7. Among the 17 cases of T1, low grade:high grade, 3:14, and all 10 cases of Tis were high grade.

The expression of heparanase is diffusely expressed in both the cell membrane and cytoplasm. The expression of heparanase protein was approximately 10% in normal urothelium but increased to approximately 30% in urothelial carcinoma samples (*p* < 0.05). Heparanase expression was elevated in high-grade compared to low-grade carcinoma samples (34.7% vs. 23.4%, respectively) (Figure 1a,b). The immunohistochemical staining of surgically resected specimens from 47 bladder cancer patients showed that positive heparanase expression was observed predominantly in cases exhibiting intravesical relapse (*p* < 0.05) (Figure 1c).

### 2.2. Knockdown of Heparanase-Induced Apoptosis in Urothelial Carcinoma Cells

Heparanase expression was studied in the human urothelial cancer cell lines MGH-U3 and T24 and found to increase compared to the normal urothelial cell line (UROtsa). The expression levels of heparanase were similar in MGH-U3 and T24 (Appendix A). We first examined the suppression of heparanase protein expression and mRNA expression by knockdown with Si RNA (Appendix A). MGH-U3 showed a significant decrease in cell activity due to heparanase knockdown compared to T24. There is a difference that MGH-U3 cells are suppressed by about 15% and T24 cells are suppressed by about 25% by knockdown by Si RNA. Inhibiting the expression of heparanase by siRNA suppressed the proliferative activity of cancer cells strongly, and cytotoxicity was observed (Figure 2a). In the MGH-U3 cell line, proliferation activity was suppressed by approximately 80% compared to approximately 40% in T24 cells. In the UROtsa cell line, heparanase knockdown suppressed growth activity by 15%. Further, heparanase knock-down by siRNA induced apoptosis (Figure 2b).

### 2.3. The Multi Enzyme Inhibitor RK-682, Which Is Also a Heparanase Inhibitor, Suppresses Cell Proliferation and Autophagy in Human Urothelial Cancer Cell Lines

RK-682 is an inhibitor of various enzymes including heparanase, phospholipase A_2, HIV-1 protease, some dual-specificity phosphatases (DSP), and a protein tyrosine phosphatase (PTP), CD45. The inhibition of heparanase by RK-682 was examined using MGH-U3 and T24 cell lines. Treatment with RK-682 suppressed heparanase protein expression and mRNA expression in these cells (Appendix A). MGH-U3 and T24 cell lines were treated with RK-682 and examined in a cell viability assay to determine cytotoxicity. RK-682-treated MGH-U3 and T24 cells showed a concentration-dependent cytotoxicity (Figure 3a). The half-maximal inhibitory concentration (IC50) of RK-682 was 78.2 nM in MGH-U3 cells, 43.2 nM in T24 cells, and 145 nM in UROtsa. The cytotoxicity was 2–3 times higher than that of the cancer cell line. UROtsa, which has low expression of heparanae, has an IC50 of RK-682 about 2–3 times higher than that of urothelial carcinoma cell line, MGH-U3 cell line and T24 cell line which has high expression of heparanase. In the heparanse knockdown experiment with Si RNA, the UROtsa cell line showed almost no inhibition of the cell activity, whereas RK-682 inhibited the cell activity of the UROtsa cell line at a high concentration. From this fact, it is considered that RK-682 has an action other than the inhibition of heparanase. The effect of RK-682 treatment was also examined in an invasion migration assay. In MGH-U3 and T24 cell lines, migratory ability decreased by approximately 20% following treatment with RK-682. In RK-682-treated MGH-U3 and T24 cells, invasion ability reduced by 55% and 40%, respectively (Figure 3b). MGH-U3 and T24 cells treated with RK-682 were also tested in an autophagy assay. The expression of autophagy decreased following the treatment with RK-682 (Figure 3c).

### 2.4. In Vivo Growth of Urothelial Carcinoma Is Suppressed by RK-682 in the BBN-Induced Mouse Bladder Cancer Model

The effects of the multi enzyme inhibitor RK-682 were tested using an in vivo model of bladder cancer treated with BBN (Figure 4a). Figure 4a briefly illustrates the experimental protocol for the present study. Six-week-old C57BL/6J mice were orally administered with 0.05% BBN. Bladder cancer-induced mice were prepared in approximately 22 weeks. The animals were divided into two groups, the treatment group (*n* = 7), and the non-treatment group (*n* = 5), totaling *n* = 14 animals. The heparanase inhibitor RK-682 was administered at 2.5 mg/100 μL, into the bladder four times a week through a catheter inserted into the bladder.

There was no significant difference in body weight between the RK-682-treated group and the control group after the end of treatment. (RK-682 group vs. control group; 24.5 g vs. 25.6 g *p* = 0.765). There was a significant decrease in the bladder weight/body wight, (RK-682 group vs. control group; 0.028 vs. 0.056 *p* = 0.0051) and ratio of infiltrative bladder cancer (RK-682 group vs. control group; 28.6% vs. 85.7% *p* = 0.027) in the RK-682 treatment group compared to the control group (Figure 4b). A TUNEL assay showed that apoptosis was more frequent in the RK-682 treatment group (Figure 4c). IHC studies of the mouse tissue showed decreased expression of Ki67, LC3, and CD31 markers in the specimens from animals treated with RK-682. In contrast, E-cadherin expression level increased after treatment with the inhibitor (Figure 5).

## 3. Discussion

Heparanase expression is elevated in many types of tumors and is associated with more aggressive cancer and a poor prognosis [3,13,14,26]. We have previously reported that syndecan-1 is involved in urothelial cancer development through the promotion of angiogenesis [10]. In this study, we investigated heparanase, an activator of syndecan-1, and analyzed its function in urothelial cancer. We have demonstrated that inhibition of heparanase suppresses cell proliferation, epithelial-mesenchymal transition (EMT), autophagy and angiogenesis.

In other cancer types, heparanase activation has been indicated in promoting metastasis and tumor progression. In bladder cancer, Gohji et al. [32] reported that cancer-specific survival rates are significantly lower when heparanase expression is elevated in bladder cancer patients. In our study, increased expression of heparanase in bladder cancer tissue samples correlated with higher recurrence rates within the bladder and progression to muscle-invasive cancer.

Chen et al. [30] suggested an association between heparanase expression and cell adhesion, and metastasis in hepatocellular carcinoma cell lines. Heparanase plays a proadhesive role in cell adhesion and tumor microembolus in hepatocellular carcinoma. In this study, inhibition of heparanase activity significantly reduced the ability of cancer cells to migrate and infiltrate. Furthermore, it was confirmed that the inhibition of autophagy resulted from heparanase inhibition.

Shteingauz et al. [33] reported the regulation of autophagy in normal and malignant cells by heparanse, conferring survival advantages and the development of resistance to chemotherapy. In the spontaneous bladder cancer mouse model, heparanase inhibition significantly suppressed bladder cancer invasion. This study confirmed that the suppression of heparanase induces apoptosis, suppresses cell proliferation and inhibits autophagy.

This report is the first to investigate heparanase inhibition and its effects on the suppression of cancer invasion, autophagy and apoptosis in bladder cancer. Intravesical treatment with a heparanase inhibitor did not result in serious side effects in the in vivo mouse model used here and hence this study was conducted as planned. As a treatment option for bladder cancer, intravesical infusion therapy allows the penetration of a drug directly into cancer cells and has mild side effects. In combination, these observations indicate that heparanase is a potential candidate for targeted therapy in bladder cancer.

This study has limitations. RK-682 is a multi-enzyme inhibitory locus targeting several enzymes including heparanase, protein tyrosine phosphatase (PTP), phospholipase A_2 and other enzymes. The T24 cell line is found to be less sensitive to siRNA inhibition compared with the MGH-U3 cell line, but is more sensitive to RK-682 inhibition. This may explain other effects of RK-682-mediated inhibition observed in this study. In UROtsa cells, the cell activity induced by siRNA was reduced by approximately 20%. However, RK-682 treatment showed cytotoxicity in a dose-dependent manner. The cytotoxicity observed was 2–3 times higher compared to the cancer cell line. Expression analysis shows that RK-682 does have a heparanase inhibitory effect (Appendix A). However, heparanase activity alone does explain changes in migration, invasion, and autophagy in bladder cancer cells. Further experiments are required to investigate the molecular function of heparanase.

## 4. Materials and Methods

### 4.1. Cell Culture, Plasmids and Chemicals

Human urothelial carcinoma cell lines MGH-U3, T24, and human urothelial cell line (UROtsa) were supplied by American Type Culture Collection (Manassas, VA, USA). MGH-U3 and T24 cells originated from human papillary bladder cancer [34]. T24 cells were cultured in RPMI1640 media supplemented with 10% fetal bovine serum and 50 units/mL penicillin-streptomycin at 37 °C in 5% CO_2_.

The antibodies, anti-Ki67, LC3, and CD31, and E-cadherin, were purchased from Abcam (Cambridge, UK). The heparanase inhibitor RK-682 was purchased from Cayman Chemical Company (Ann Arbor, MI, USA). siRNA molecules were purchased from Thermo Fisher Scientific (K.K. Japan).

### 4.2. siRNA Transfection of Heparanase

For transfection analyses, 10^6^ cells from each cell line were seeded into 6-cm dishes. They were transfected with either 100 nM of control RNA (Santa Cruz Biotechnology, Dallas, TX, USA) or with the heparanase siRNA. Transfections were performed with the Lipofectamine system (Invitrogen Japan, Tokyo, Japan) following the manufacturer’s protocol.

The primers used were: HPSE sense 5′-AGUACUUGCGGUUACCCUATT-3′; HPSE antisense 5′-UAGGGUAACCGCAAGUACUTG-3′.

Actin sense 5′-CTCTTCCAGCCTTCCTTCCT-3′; Actin antisense 5′-AGCACTGTGTTGGCGTACAG-3′.

Gene expression analysis of cell cycle-related genes was performed by qPCR using the PrimerArray Cell Cycle (Takara, Otsu, Japan).

### 4.3. Tissue Samples and Immunohistochemistry (IHC)

This study was approved by the Medical Ethics Committee of Nara Medical University. The requirement for informed patient consent was waived due to the retrospective nature of the study (Study ID: NMU 900, July 23, 2013). A total of 57 patients diagnosed with organ-confined urothelial cancer between April 2007 and June 2010 at the Nara Medical University hospital were included in this study. The clinicopathological data and follow-up data were collected via a retrospective chart review. All pathological examinations were performed under the guidance of two pathologists (K.S. and N.K.) according to the 2009 TNM classification system [32]. All patients had bladder cancer and underwent transurethral resection of the bladder tumor (TURBT). IHC staining of 57 TURBT specimens using paraffin-embedded, formalin-fixed tissue blocks was performed as previously described [10,33]. Antibodies against heparinase were used as the primary antibodies at a dilution of 1:500. Staining was scored based on the positive cell ratio using standard light microscopy. Staining outcomes were evaluated by two independent observers (Y.T. and K.S), who were blinded to patient clinicopathological data.

### 4.4. Cell Proliferation Assay

The CellTiter 96 AQueous One Solution Cell Proliferation Assay (Promega, Madison, WI, USA) was used to measure cell proliferation by MTS assay as previously described [11]. Data were collected from triplicate experiments.

### 4.5. Apoptosis Detection Assay

Following the transfection with siRNA, cells were stained with propidium iodide (PI) and Fluorescein-5-isothiocyanate (FITC)-conjugated Annexin V (AV) following the manufacturer’s protocol (TACS Annexin V-FITC kit; R&D Systems). Apoptotic cells were quantified by calculating the number of cells positive for AV and negative for PI. Experiments were performed a minimum three times of duplicate.

### 4.6. TdT-Mediated dUTP Nick End Labeling (TUNEL) Assay

Formalin-fixed and paraffin-embedded 5-l-m thick sections of tumor specimens were stained using a TUNEL assay: Tumor TACS in situ apoptosis detection kit (R&D Systems, Minneapolis, MN, USA). The apoptotic index (the number of apoptotic cells per total number of cells) was calculated as per 400 microscopic fields per sample.

### 4.7. Cell Viability Assay

UROtsa, MGH-U3 and T24 cells were seeded into 96-well plates at 2 × 10^3^ cells/well and incubated overnight. The growth medium was removed, and the cells were washed once with phosphate-buffered saline (PBS). Fresh serum-free medium with or without heparinase inhibitor (RK-682) was applied. Cells were then treated with RK-682 (between 1–1000 nM) for 72 h to evaluate cell viability. The IC50 was determined based on the concentration-effect relationship using PRISM software version 5.00 (GraphPad Software, San Diego, CA, USA). Cell viability was measured using a Cell Counting Kit-8 (Dojindo, Kumamoto, Japan) according to the manufacturer’s protocol. Absorbance was measured at 490 nm with a reference at 630 nm using an Infinite 200M PRO microplate autoreader (Tecan, Männedorf, Switzerland).

### 4.8. Migration Assay

A migration assay was performed using the BD Falcon FluoroBlok Insert System (Becton Dickinson, Franklin Lakes, NJ, USA) according to the manufacturer’s instructions. Cells were starved in serum-free media for 24 h, then seeded at a density of 2.5 × 10^4^ cells/well in serum-free media plus RK-682 inhibitor. RPMI1640 media with 10% FBS chemoattractant was contained within the lower wells. The cells were incubated in a humidified environment at 37 °C with 5% CO_2_ for 48 h. Cells that were attached to the membrane were stained with the cell viability indicator Calcein AM Fluorescent Dye for 30 min (Promo Kine, Heidelberg, Germany) and quantified with an Infinite 200M PRO microplate spectrophotometer (Tecan, Männedorf, Switzerland) at 495 mm excitation and 515 nm emission. Cells were inspected via fluorescent microscopy.

### 4.9. Autophagy Assay

Autophagy assays were performed using Muse™ Cell Analyzer from Millipore (Hayward, CA, USA) following the manufacturer’s instructions. Following treatment of MGH-U3 and T24 cells with heparinase inhibiter RK-682 (Cayman Chemical Company, Ann Arbor, MI, USA), the treated cells were washed with PBS buffer. The autophagy assays were analyzed using the Muse™ Autophagy LC3-antibody based kit (Millipore) according to the manufacturer’s protocol.

### 4.10. BBN-Induced Mouse Bladder Cancer Model

An in vivo mouse model of bladder cancer was treated with BBN. Fourteen 6-week-old C57BL/6J female mice were obtained from Oriental Bio Service (Kyoto, Japan). All animal studies were approved by Nara Medical University affiliated Frist People’s Hospital Committee on Use and Care of Animals and conducted in accordance with local humane animal care standard (Reference Number: 11389). BBN B0938 (Tokyo Chemical Industry, Tokyo, Japan) treated bladder cancer model mice were treated with the heparanase inhibitor RK-682 in the bladder once a week for four weeks by injection. Mice were randomized into a control group (control PBS, *n* = 7) or heparanase inhibitor (RK-682, *n* = 7) treatment group and received a single intravesical treatment instillation that was retained for 1 h with a purse-string suture. Mice were sacrificed (at week 25) after four treatments. Bladders were removed, placed open on filter paper, and fixed in 10% neutral buffered formalin. Bladders were then embedded in paraffin, step-sectioned, and stained with hematoxylin-eosin (H&E) and IHC staining for E-cadherin, Ki67, CD31 and LC3 (Abcam, Tokyo, Japan).

### 4.11. Statistical Analysis

Differences in cell migration were evaluated with the Student’s *t*-test. The correlation of IHC staining intensity was assessed with the Man–Whitney U test. IBM SPSS Version 21 (SPSS Inc., Chicago, IL, USA) and PRISM software version 5.00 (San Diego, CA, USA) were used for statistical analyses and data plotting, respectively. Statistical significance was set at *p* < 0.05, and all reported *p* values were two-sided.

## 5. Conclusions

Heparanase induces invasion and autophagy in urothelial carcinoma. Downregulation of heparanase induces apoptosis. Heparanase may contribute to urothelial carcinoma cell survival and invasion.

## Figures and Tables

**Figure 1 ijms-21-03789-f001:**
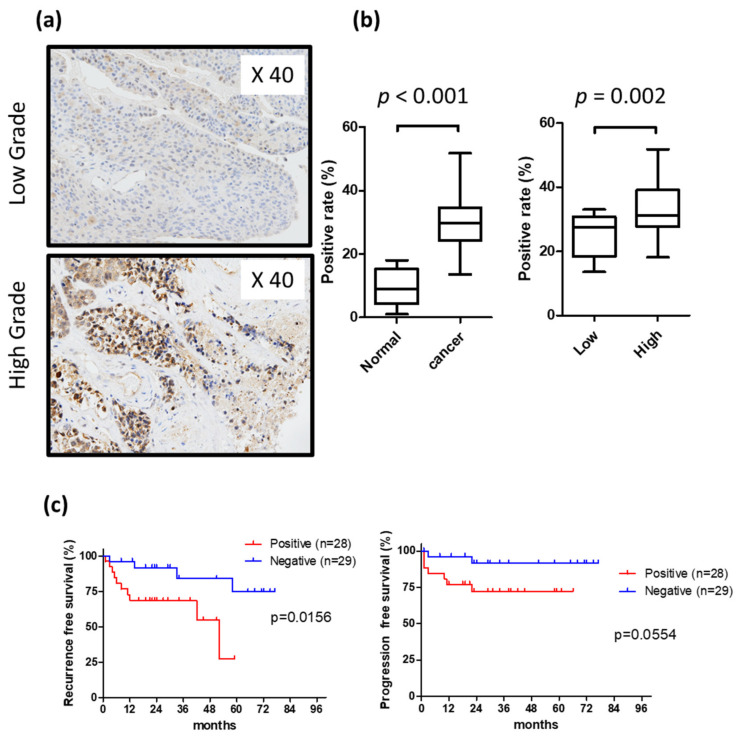
Immunohistological examination of expression of heparanase in bladder tissue; (**a**) positive ratio in low grade bladder cancer and high grade bladder cancer; (**b**) heparanase expression rate; (**c**) Kaplan–Meier curve of intravesical recurrence and invasion.

**Figure 2 ijms-21-03789-f002:**
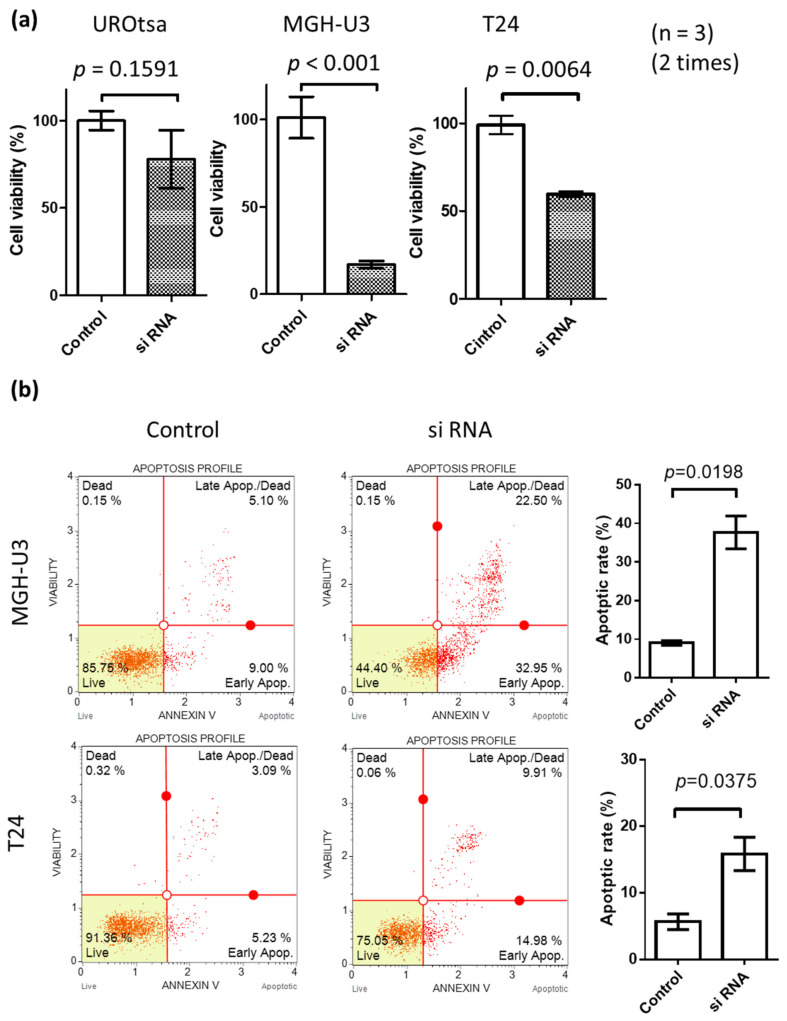
(**a**) Effect of heparanase knockdown on cell survival in urothelial carcinoma cells. Cell viability was assessed by an MTS assay 72 h following transfection; (**b**) 48 h following transfection, cells stained with Annexin V and propidium iodide were analyzed by flow cytometry (upper panels) and the percentages of apoptotic cells (AV[+]/PI[)]) calculated (lower panels). Inset photograph is an immunofluorescence microscopy image showing cells positive for FITC-conjugated Annexin V (AV). Each value is the mean ± standard error. C, control RNA (non-specific siRNA); Si RNA, heparanase siRNA.

**Figure 3 ijms-21-03789-f003:**
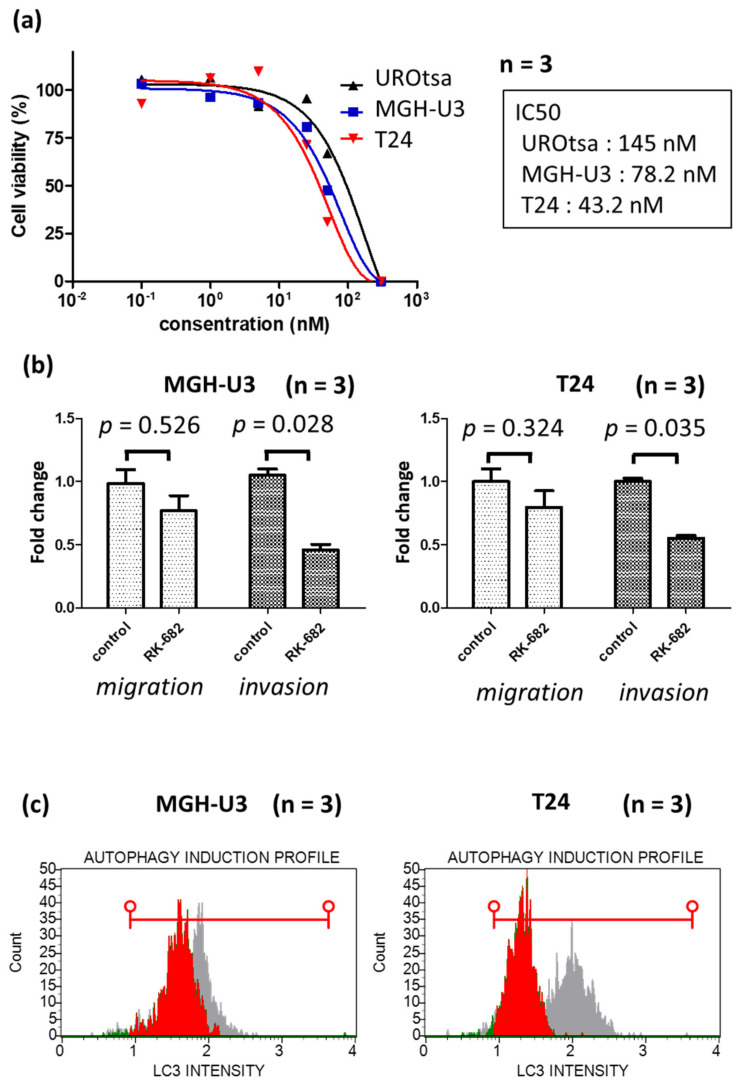
(**a**) Treatment with the multi enzyme inhibitor RK-682 inhibited cell proliferation in MGH-U3, T24 cells and UROtsa. Cells were incubated in serum-free media for 24 h and treated with different concentrations of RK-682 for a further 48 h. The number of viable cells was measured by an MTS assay and expressed as a percentage of viable cells; (**b**) effect of RK-682 on MGH-U3 and T24 cells. RK-682 treatment resulted in a significant inhibition of MGH-U3 and T24 cell invasion (*p* < 0.05); (**c**) 48 h treatment with heparanase inhibitor RK-682 inhibited cell autophagy in MGH-U3 and T24 cells. The red horizontal line shows the range of LC3 intensity after KR-682 treatment.

**Figure 4 ijms-21-03789-f004:**
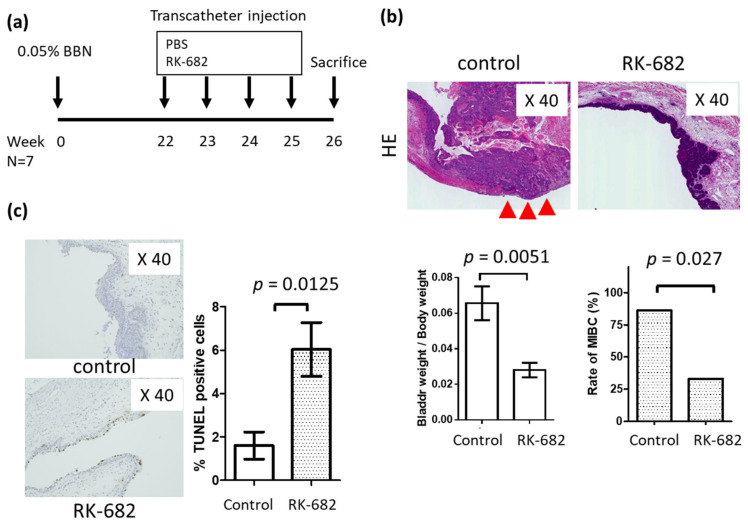
Intravesical injection of RK-682 inhibits in vivo tumor growth in the mouse N-butyl-N-(4-hydroxybutyl) nitrosamine (BBN)-induced bladder cancer implant model. (**a**) Diagrammatic experimental procedure; (**b**) RK-682 or control was transurethrally instilled into the bladder lumen. Bladders were resected post-instillation. Hematoxylin Eosin (HE) staining of bladder, comparison of bladder weight, ratio of muscle layer infiltration. Red triangle indicate images of muscle invasive bladder cancer.; (**c**) the percentage of cells in resected bladder specimens immunoreactive with TUNEL reagent, calculated per 1000 cells/in a high-power field. Each value is the mean ± standard error.

**Figure 5 ijms-21-03789-f005:**
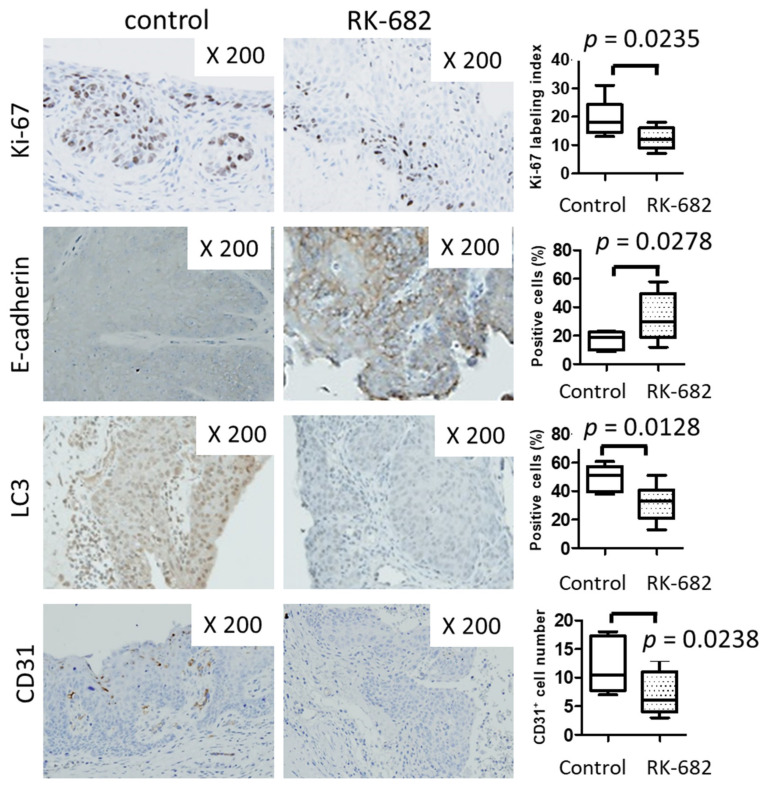
Immunohistochemical analysis of the mouse BBN-induced bladder cancer implant model. Immunohistochemistry for Ki-67, E-cadherin, LC3 and CD31 expression. The percentage of immunopositive cells was determined per 1000 cells in a high-power field.

**Table 1 ijms-21-03789-t001:** Characterization of urothelial carcinomas.

	*p*Ta (*n* = 20)	*p*T1 (*n* = 17)	*p*Tis (*n* = 10)
Age	71.3 (61–82)	72.9 (5–80)	72.4 (62–86)
Gender (M:F)	16:4	14:3	8:2
Grade			
Low grade	13	3	0
High grade	7	14	10

*p*Ta = low-grade non-muscle invasive bladder cancer; *p*T1 = intermediate risk non-muscle invasive bladder cancer; *p*Tis = in situ neoplasia.

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
