# Peer review of "Inhibition of Heparanase Expression Results in Suppression of Invasion, Migration and Adhesion Abilities of Bladder Cancer Cells"

_ijms, 2020, doi:10.3390/ijms21113789_

Round 1
Reviewer 1 Report
This work intends to demonstrate that the enzyme heparinase (which activates syndecan-1) can be targeted to restrict proliferation of urothelial carcinoma cells and to reduce urothelial tumor growth in mice. The topic is interesting; experiments are well described and the data are solid and varied (in vitro + in vivo + clinical samples). The writing is excessively concise.
There are important issues with this Ms:
Fig 2. MGH-U3 cells are much more sensitive to siRNA-heparanase that T24 cells? Explanation? Do these cells express different levels of heparanase?
Fig 3a presents the growth inhibition curve for three cell lines (normal and 2 cancer lines) but the text only refers to the cancer cell lines, omitting normal cells (IC50?) also sensitive to the drug. The choice of RK-682 as a heparinase inhibitor is highly objectionable because this drug is NOT only a herapanase inhibitor but is known to function as a tyrosine phosphatase inhibitor, therefore inhibiting multiple enzymes (see Carneiro et al., Eur J Med Chem 2015;97:42). Therefore, it is not wise to interpret the Results in terms of heparinase inhibition. This part must be significantly revised. The multi-target activity of RK-682 likely explains why T24 cells are much more sensitive to this compound than the siRNA (Fig1 vs Fig2).
The in vivo data are OK, but here again, the data show the anticancer activity of RK-682, which is NOT a selective inhibitor of heparanase.
It is unfair (and not professional) to cite RK-682 as a heparinase inhibitor, whereas the literature largely mentions this drug as a broad PTP1 inhibitor.
The abstract refers to angiogenesis, but there is no data to support this activity. Delete
Overall, this is a potentially interesting study but the presentation of the data can be improved and most importantly, the Results and Discussion parts must be profoundly rewritten to refer to the multiple activities of RK-682 and its lack of selectivity for heparanase.
Other specific comments:
Legend to Table 1: explain to what correspond the grades pT1, pTa and pTis, otherwise it is useless.
Fig 1. Detail further the expression of heparinase examined by IHC. Only membrane expression and/or cytoplasmic expression? The IHC photos are OK but it would be useful to present other photos at higher magnitude, to define the type of cell staining. Indicate the type of staining (punctate, uniform, diffuse?).
Line 96: indicate “cancer” urothelial cell lines
Line 99-100: MGH-U3 mentioned twice
Line 156 and following: “We have previously shown….” References are lacking.
Author Response
Response to Reviewer’s Comments
Reviewer 1:
Comments and Suggestions for Authors
This work intends to demonstrate that the enzyme heparinase (which activates syndecan-1) can be targeted to restrict proliferation of urothelial carcinoma cells and to reduce urothelial tumor growth in mice. The topic is interesting; experiments are well described and the data are solid and varied (in vitro + in vivo + clinical samples). The writing is excessively concise.
There are important issues with this Ms:
Fig 2. MGH-U3 cells are much more sensitive to siRNA-heparanase that T24 cells? Explanation? Do these cells express different levels of heparanase?
Reply: Thank you for your comment. The expression of heparanase was the same in MGH-U3 cells and T24 cells at the protein level and mRNA level, respectively. However, there is a difference between the two cell lines in that expression in MGH-U3 cells is suppressed by approximately 15% compared to approximately 25% in T24 cells by siRNA knockdown. These results are shown in an additional Figure (Supplementary Figure 2).
Fig 3a presents the growth inhibition curve for three cell lines (normal and 2 cancer lines) but the text only refers to the cancer cell lines, omitting normal cells (IC50?) also sensitive to the drug.
Reply: Thank you for your comment. The following text was added to line 178:
“, and 145 nM in UROtsa.”
The choice of RK-682 as a heparinase inhibitor is highly objectionable because this drug is NOT only a herapanase inhibitor but is known to function as a tyrosine phosphatase inhibitor, therefore inhibiting multiple enzymes (see Carneiro et al., Eur J Med Chem 2015;97:42). Therefore, it is not wise to interpret the Results in terms of heparinase inhibition. This part must be significantly revised. The multi-target activity of RK-682 likely explains why T24 cells are much more sensitive to this compound than the siRNA (Fig1 vs Fig2).
The in vivo data are OK, but here again, the data show the anticancer activity of RK-682, which is NOT a selective inhibitor of heparanase.
It is unfair (and not professional) to cite RK-682 as a heparinase inhibitor, whereas the literature largely mentions this drug as a broad PTP1 inhibitor.
Reply: Thank you for your comment. The manuscript has been modified to reflect these points.
In Section 2.3., the following material has been added to clarify the use of RK-682:
“The multi enzyme inhibitor RK-682, which is also a heparanase inhibitor, suppresses cell proliferation and autophagy in human urothelial cancer cell lines
RK-682 is an inhibitor of various enzymes including heparanase, phospholipase A_2, HIV-1 protease, some dual-specificity phosphatases (DSP), and a protein tyrosine phosphatase (PTP), CD45. The inhibition of heparanase by RK-682 was examined using MGH-U3 and T24 cell lines. Treatment with RK-682 suppressed heparanase protein expression and mRNA expression in these cells. (Supplementary Figure 3).”
In the discussion section, the following text has been added to lines 395 – 405:
“This study has limitations. RK-682 is a multi-enzyme inhibitory locus targeting several enzymes including heparanase, protein tyrosine phosphatase (PTP), phospholipase A_2 and other enzymes. The T24 cell line is found to be less sensitive to siRNA inhibition compared with the MGH-U3 cell line, but is more sensitive to RK-682 inhibition. This may explain other effects of RK-682-mediated inhibition observed in this study. In UROtsa cells, the cell activity induced by siRNA was reduced by approximately 20%. However, RK-682 treatment showed cytotoxicity in a dose-dependent manner. The cytotoxicity observed was 2-3 times higher compared to the cancer cell line. Expression analysis shows that RK-682 does have a heparanase inhibitory effect (Supplementary Figure 3). However, heparanase activity alone does explain changes in migration, invasion, and autophagy in bladder cancer cells. Further experiments are required to investigate the molecular function of heparanase.”
The abstract refers to angiogenesis, but there is no data to support this activity. Delete
Reply: Thank you for your comment. The reference to angiogenesis within the abstract has been deleted.
Overall, this is a potentially interesting study but the presentation of the data can be improved and most importantly, the Results and Discussion parts must be profoundly rewritten to refer to the multiple activities of RK-682 and its lack of selectivity for heparanase.
Other specific comments:
Legend to Table 1: explain to what correspond the grades pT1, pTa and pTis, otherwise it is useless.
Reply: Thank you for your comment. A legend describing each term is placed below Table 1.
Fig 1. Detail further the expression of heparinase examined by IHC. Only membrane expression and/or cytoplasmic expression? The IHC photos are OK but it would be useful to present other photos at higher magnitude, to define the type of cell staining. Indicate the type of staining (punctate, uniform, diffuse?).
Reply: the following sentence describing the staining pattern has been added to line 128:
“The expression of heparanase is diffusely expressed in both the cell membrane and cytoplasm”
The images shown in Figure 1 are the highest resolution captured in the experiments.
Line 96: indicate “cancer” urothelial cell lines
Reply: Thank you for your comment. The word cancer has been added to the sentence. This is now line 140.
Line 99-100: MGH-U3 mentioned twice
Reply: Thank you for your comment. The word duplication has been removed from the sentence.
Line 156 and following: “We have previously shown….” References are lacking.
Reply: Thank you for your comment. The relevant reference has been added.
Reviewer 2 Report
Heparanase is an endoglucuronidase that cleaves heparan sulfate chains of proteoglycans. Increased expression of heparinase correlates with an aggressive tumor phenotype. A major consequence of heparanase action in cancer is a robust up-regulation of growth factor expression and increased shedding of syndecan-1 (a transmembrane heparan sulfate proteoglycan, CD138). Preclinical experiments have found that heparanase inhibitors to substantially reduce tumor growth and metastasis. The manuscript by Tatsumi et al report that inhibition of heparanase expression results in suppression of invasion, migration, and adhesion abilities of bladder cancer cells. The manuscript is well written, the first report of heparanase inhibition in bladder cancer cells in vitroand ex vivo, the data support the authors' general assumptions and conclusions. However, the mechanisms underlying the effects on phenotype in bladder cancer cells by heparanase inhibition need works to clarify.
Major comments:
- The knockdown heparanase expression by siRNA in two urothelial carcinoma cell lines, MGH-U3 and T24, leads to suppress the proliferation and induce apoptosis, but the study did not present the level changes of heparanase expression in mRNA and protein.
- RK-682 inhibits several protein tyrosine phosphatases, which are keys to control cell growth and proliferation, differentiation, and survival or apoptosis. The authors need comment and clarify this consideration. Does RK-682, such different analogous or derivative, can selective heparanase activity specify? Or does RK-682 can reduce the activity of heparanase in treated cells and tumor tissues?
Minor Comments:
Authors need to add samples numbers and p-values on some figures, and figure legends, as indicated in detail below.
- In figure 2. A. Need show experiment sample numbers and how many independent experiments was repeated. B. In (b), should show bar figure also, indicated how many samples and independent experiments had been done. C. why it has more late apoptosis/dead cells in T24 control group? need clarify its result. D. is control group is with nonspecific siRNA control?
- In figure 3. A. in (a), IC50 of RK-682 is significantly different among MGH-U3, T24 and UROtsa cell lines? B. P values and samples numbers need label in (b) and (c).
- In figure 4. A. P values and samples numbers need label in (b) and (c). B. Dose RK-682 affect on body weight after transcatheter injection for 5 weeks? If it is yes, need calculate bladder weight per body weight.
Author Response
Response to Reviewer’s Comments
Reviewer 2:
Heparanase is an endoglucuronidase that cleaves heparan sulfate chains of proteoglycans. Increased expression of heparinase correlates with an aggressive tumor phenotype. A major consequence of heparanase action in cancer is a robust up-regulation of growth factor expression and increased shedding of syndecan-1 (a transmembrane heparan sulfate proteoglycan, CD138). Preclinical experiments have found that heparanase inhibitors to substantially reduce tumor growth and metastasis. The manuscript by Tatsumi et al report that inhibition of heparanase expression results in suppression of invasion, migration, and adhesion abilities of bladder cancer cells. The manuscript is well written, the first report of heparanase inhibition in bladder cancer cells in vitro and ex vivo, the data support the authors' general assumptions and conclusions. However, the mechanisms underlying the effects on phenotype in bladder cancer cells by heparanase inhibition need works to clarify.
Major comments:
- The knockdown heparanase expression by siRNA in two urothelial carcinoma cell lines, MGH-U3 and T24, leads to suppress the proliferation and induce apoptosis, but the study did not present the level changes of heparanase expression in mRNA and protein.
Reply: Thank you for your comment. The following sentence was added to section 2.3. in line 173-175:
“Treatment with RK-682 suppressed heparanase protein expression and mRNA expression in these cells. (Supplementary Figure 3).”
Supplementary Figure 3 shows western blots and bar charts of cells treated with RK-682 demonstrating reduced expression of heparanase.
- RK-682 inhibits several protein tyrosine phosphatases, which are keys to control cell growth and proliferation, differentiation, and survival or apoptosis. The authors need comment and clarify this consideration. Does RK-682, such different analogous or derivative, can selective heparanase activity specify? Or does RK-682 can reduce the activity of heparanase in treated cells and tumor tissues?
Reply: Thank you for your comment. The manuscript has been modified to reflect these points.
In Section 2.3., the following material has been added to clarify the use of RK-682:
“The multi enzyme inhibitor RK-682, which is also a heparanase inhibitor, suppresses cell proliferation and autophagy in human urothelial cancer cell lines
RK-682 is an inhibitor of various enzymes including heparanase, phospholipase A_2, HIV-1 protease, some dual-specificity phosphatases (DSP), and a protein tyrosine phosphatase (PTP), CD45. The inhibition of heparanase by RK-682 was examined using MGH-U3 and T24 cell lines. Treatment with RK-682 suppressed heparanase protein expression and mRNA expression in these cells. (Supplementary Figure 3).”
In the discussion section, the following text has been added to lines 395 – 405:
“This study has limitations. RK-682 is a multi-enzyme inhibitory locus targeting several enzymes including heparanase, protein tyrosine phosphatase (PTP), phospholipase A_2 and other enzymes. The T24 cell line is found to be less sensitive to siRNA inhibition compared with the MGH-U3 cell line, but is more sensitive to RK-682 inhibition. This may explain other effects of RK-682-mediated inhibition observed in this study. In UROtsa cells, the cell activity induced by siRNA was reduced by approximately 20%. However, RK-682 treatment showed cytotoxicity in a dose-dependent manner. The cytotoxicity observed was 2-3 times higher compared to the cancer cell line. Expression analysis shows that RK-682 does have a heparanase inhibitory effect (Supplementary Figure 3). However, heparanase activity alone does explain changes in migration, invasion, and autophagy in bladder cancer cells. Further experiments are required to investigate the molecular function of heparanase.”
Minor Comments:
Authors need to add samples numbers and p-values on some figures, and figure legends, as indicated in detail below.
- In figure 2.
- Need show experiment sample numbers and how many independent experiments was repeated.
Reply: Thank you for your comment. The sample numbers (n = 3) and replications (twice) have been added to Figure 2.
- In (b), should show bar figure also, indicated how many samples and independent experiments had been done.
Reply: Thank you for your comment. Bar charts have been included in Figure 2b
- why it has more late apoptosis/dead cells in T24 control group? need clarify its result.
Reply: thank you for your comment. This was a mistake in the Figure. It has been corrected. In the correct Figure, late apoptosis / dead cells are 3%, which is similar to the MGH-U3 cells.
- is control group is with nonspecific siRNA control?
Reply: Thank you for your comment. Clarification has been added to the Figure.
- In figure 3.
- in (a), IC50 of RK-682 is significantly different among MGH-U3, T24 and UROtsa cell lines?
Reply: Thank you for your comment. UROtsa, which has low expression of heparanase, has an RK-682 IC50 approximately 2-3 times higher than that of the urothelial carcinoma cell lines, MGH-U3 cell line and T24 cell line (which both exhibit higher expression levels of heparanase). In the heparanase siRNA knockdown experiment, the UROtsa cell line showed almost no inhibition of cell activity, whereas RK-682 inhibited the cell activity of the UROtsa cell line at a high concentration. From this data, it is possible that RK-682 has one or more actions apart from the inhibition of heparanase.
- P values and samples numbers need label in (b) and (c).
Reply: the missing p values and sample numbers have been added.
- In figure 4. A. P values and samples numbers need label in (b) and (c). B. Dose RK-682 affect on body weight after transcatheter injection for 5 weeks? If it is yes, need calculate bladder weight per body weight.
Reply: the missing p values and sample numbers have been added.
There was no significant difference in body weight between the RK-682-treated group and the control group at the end of treatment (RK-682 group v.s. control group; 24.5g v.s. 25.6g, p = 0.765). There was a significant decrease in the bladder weight / body wight (RK-682 group v.s. control group; 0.028 v.s. 0.056, p = 0.0051), and ratio of infiltrative bladder cancer (RK-682 group v.s. control group; 28.6% v.s. 85.7%, p = 0.027) in the RK-682 treatment group compared to the control group (Figure 4b).
Round 2
Reviewer 1 Report
All my previous comments have been properly adressed. I can now recommend the publication of the revised manuscript. This is a good research work, useful and interesting.
Reviewer 2 Report
The manuscript by Tatsumi et al report beneficial effects in suppression of invasion, migration, and adhesion in bladder cancer cells by inhibition of heparanase expression. The authors have addressed my comments and answered my questions and consideration. The manuscript is well written, on the most part scientifically sound, and the data support the authors' general assumptions and conclusions.